# Q-Transformer: Scalable Offline Reinforcement Learning via Autoregressive Q-Functions

**Yevgen Chebotar**[*], **Quan Vuong**[*], **Alex Irpan, Karol Hausman, Fei Xia, Yao Lu, Aviral Kumar, Tianhe Yu, Alexander Herzog, Karl Pertsch, Keerthana Gopalakrishnan, Julian Ibarz, Ofir Nachum, Sumedh Sontakke, Grecia Salazar, Huong T Tran, Jodilyn Peralta, Clayton Tan, Deeksha Manjunath, Jaspiar Singht, Brianna Zitkovich, Tomas Jackson, Kanishka Rao, Chelsea Finn, Sergey Levine**

Google DeepMind

**Abstract:** In this work, we present a scalable reinforcement learning method for training multi-task policies from large offline datasets that can leverage both human demonstrations and autonomously collected data. Our method uses a Transformer to provide a scalable representation for Q-functions trained via offline temporal difference backups. We therefore refer to the method as Q-Transformer. By discretizing each action dimension and representing the Q-value of each action dimension as separate tokens, we can apply effective high-capacity sequence modeling techniques for Q-learning. We present several design decisions that enable good performance with offline RL training, and show that Q-Transformer outperforms prior offline RL algorithms and imitation learning techniques on a large diverse real-world robotic manipulation task suite. The project's website and videos can be found at qtransformer.github.io

## 1 Introduction

Robotic learning methods that incorporate large and diverse datasets in combination with high-capacity expressive models, such as Transformers [1, 2, 3, 4, 5, 6], have the potential to acquire generalizable and broadly applicable policies that perform well on a wide variety of tasks [1, 2]. For example, these policies can follow natural language instructions [4, 7], perform multi-stage behaviors [8, 9], and generalize broadly across environments, objects, and even robot morphologies [10, 3]. However, many of the recently proposed high-capacity models in the robotic learning literature are trained with supervised learning methods. As such, the performance of the resulting policy is limited by the degree to which human demonstrators can provide high-quality demonstration data. This is limiting for two reasons. First, we would like robotic systems that are *more* proficient than human teleoperators, exploiting the full potential of the hardware to perform tasks quickly, fluently, and reliably. Second, we would like robotic systems that get better with autonomously gathered experience, rather than relying entirely on high-quality demonstrations.

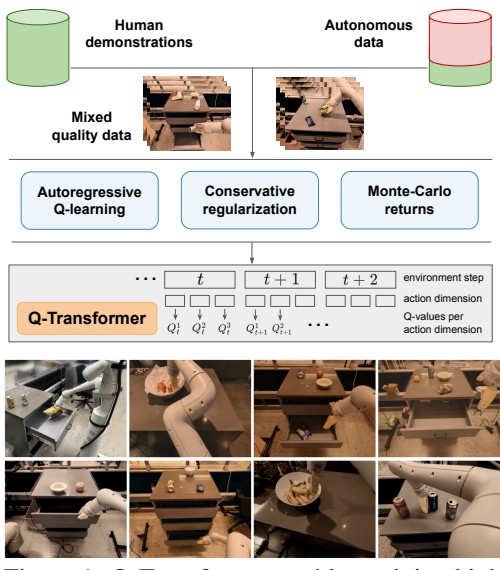

Figure 1: Q-Transformer enables training high-capacity sequential architectures on mixed quality data. Our policies are able to improve upon human demonstrations and execute a variety of manipulation tasks in the real world.

Reinforcement learning in principle provides both of these capabilities. A number of promising recent advances demonstrate the successes of large-scale robotic RL in varied settings, such as robotic grasping and stacking [11, 12], learning heterogeneous tasks with human-specified rewards [13], learning multi-task policies [14, 15], learning goal-conditioned policies [16, 17, 18, 19], and robotic navigation [20, 21, 22, 23, 24]. However,

---

[*] Equal contribution.

Corresponding emails: chebotar@google.com, quanhovuong@google.com.

7th Conference on Robot Learning (CoRL 2023), Atlanta, USA.

training high-capacity models such as Transformers using RL algorithms has proven more difficult to instantiate effectively at large scale. In this paper, we aim to combine large-scale robotic learning from diverse real-world datasets with modern high-capacity Transformer-based policy architectures.

While in principle simply replacing existing architectures (e.g., ResNets [15] or smaller convolutional neural networks [11, 14]) with a Transformer is conceptually straightforward, devising a methodology that effectively makes use of such architectures is considerably more challenging. High-capacity models only make sense when we train on large and diverse datasets – small, narrow datasets simply do not require this much capacity and do not benefit from it. While prior works used simulation to create such datasets [2, 25, 26], the most representative data comes from the real world [12, 11, 14]. Therefore, we focus on reinforcement learning methods that can use Transformers and incorporate large, previously collected datasets via *offline* RL. Offline RL methods train on prior data, aiming to derive the most effective possible policy from a given dataset. Of course, this dataset can be augmented with additionally autonomously gathered data, but the training is separated from data collection, providing an appealing workflow for large-scale robotics applications [27].

Another issue in applying Transformer models to RL is to design RL systems that can effectively train such models. Effective offline RL methods generally employ Q-function estimation via temporal difference updates [28]. Since Transformers model discrete token sequences, we convert the Q-function estimation problem into a discrete token sequence modeling problem, and devise a suitable loss function for each token in the sequence. Naïvely discretizing the action space leads to exponential blowup in action cardinality, so we employ a per-dimension discretization scheme, where each dimension of the action space is treated as a *separate time step* for RL. Different bins in the discretization corresponds to distinct actions. The per-dimension discretization scheme allows us to use simple discrete-action Q-learning methods with a conservative regularizer to handle distributional shift [29, 30]. We propose a specific regularizer that minimizes values of every action that was *not* taken in the dataset and show that our method can learn from both narrow demonstration-like data and broader data with exploration noise. Finally, we utilize a hybrid update that combines Monte Carlo and $n$-step returns with temporal difference backups [31], and show that doing so improves the performance of our Transformer-based offline RL method on large-scale robotic learning problems.

In summary, our main contribution is the Q-Transformer, a Transformer-based architecture for robotic offline reinforcement learning that makes use of per-dimension tokenization of Q-values and can readily be applied to large and diverse robotic datasets, including real-world data. We summarize the components of Q-Transformer in Figure 1. Our experimental evaluation validates the Q-Transformer by learning large-scale text-conditioned multi-task policies, both in simulation for rigorous comparisons and in large-scale real-world experiments for realistic validation. Our real-world experiments utilize a dataset with 38,000 *successful* demonstrations and 20,000 *failed* autonomously collected episodes on more than 700 tasks, gathered with a fleet of 13 robots. Q-Transformer outperforms previously proposed architectures for large-scale robotic RL [15, 14], as well as previously proposed Transformer-based models such as the Decision Transformer [32, 33].

## 2  Related Work

Offline RL has been extensively studied in recent works [34, 35, 36, 37, 35, 38, 39, 40, 41, 42, 43, 44, 45, 46, 47, 48, 39]. Conservative Q-learning (CQL) [29] learns policies constrained to a conservative lower bound of the value function. Our goal is not to develop a new algorithmic principle for offline RL, but to devise an offline RL system that can integrate with high-capacity Transformers, and scale to real-world multi-task robotic learning. We thus develop a version of CQL particularly effective for training large Transformer-based Q-functions on mixed quality data. While some works have noted that imitation learning outperforms offline RL on demonstration data [49], other works showed offline RL techniques to be effective with demonstrations both in theory and in practice [50, 15]. Nonetheless, a setting that combines "narrow" demonstration data with "broad" sub-optimal (e.g., autonomously collected) data is known to be particularly difficult [51, 52, 53], though it is quite natural in many robotic learning settings where we might want to augment a core set of demonstrations with relatively inexpensive low-quality autonomously collected data. We believe that the effectiveness of our method in this setting is of particular interest to practitioners.

Transformer-based architectures [54] have been explored in recent robotics research, both to learn generalizable task spaces [55, 56, 57, 58, 8, 59] and to learn multi-task or even multi-domain sequential policies directly [2, 1, 6, 3]. Although most of these works considered Transformers in a supervised learning setting, e.g., learning from demonstrations [4, 5], there are works on employ-

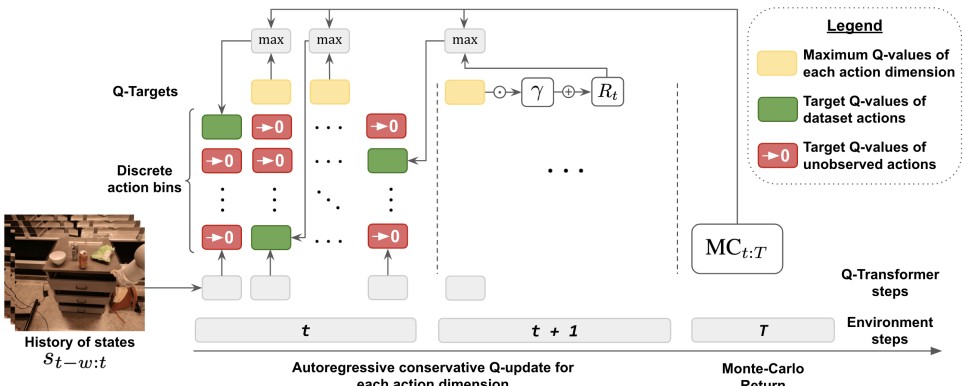

Figure 2: **Q-values update for each action dimension at timestep** $t$**.** Given a history of states, we update the Q-values of all bins in all action dimensions. The Q-values of the discrete action bins of the *dataset actions* are trained via the Bellman update (green boxes). The values of action bins not observed in the dataset are minimized towards zero (red boxes). The Q-targets of all action dimensions except the last one are computed using maximization over the next action dimension within the same time step. The Q-target of the last action dimension is computed using the discounted maximization of the first dimension of the next time step plus the reward. We also incorporate Monte Carlo returns by taking the maximum of the computed Q-targets and the return-to-go.

ing Transformers for RL and conditional imitation learning [32, 20, 60, 33]. In our experiments, we compare to Decision Transformer (DT) in particular [32], which extends conditional imitation learning with reward conditioning [61, 62] to use sequence models, and structurally resembles imitation learning methods that have been used successfully for robotic control. Although DT incorporates elements of RL (namely, reward functions), it does not provide a mechanism to improve over the demonstrated behavior or recombine parts of the dataset to synthesize more optimal behaviors, and indeed is known to have theoretical limitations [63]. On the other hand, such imitation-based recipes are popular perhaps due to the difficulty of integrating Transformer architectures with more powerful temporal difference methods (e.g., Q-learning). We show that several simple but important design decisions are needed to make this work, and our method significantly outperforms non-TD methods such as DT, as well as imitation learning, on our large-scale multi-task robotic control evaluation. Extending Decision Transformer, Yamagata et al. [64] proposed to use a Q-function in combination with a Transformer-based policy, but the Q-function itself did not use a Transformer-based architecture. Our Q-function could in principle be combined with this method, but our focus is specifically on directly training Transformers to represent Q-values.

To develop a Transformer-based Q-learning method, we discretize each action space dimension, with each dimension acting as a distinct time step. Autoregressive generation of discrete actions has been explored by Metz et al. [65], who propose a hierarchical decomposition of an MDP and then utilize LSTM [66] for autoregressive discretization. Our discretization scheme is similar but simpler, in that we do not use any hierarchical decomposition but simply treat each dimension as a time step. However, since our goal is to perform offline RL at scale with real-world image based tasks (vs. the smaller state-space tasks learned via online RL by Metz et al. [65]), we present a number of additional design decisions to impose a conservative regularizer, enabling training our Transformer-based offline Q-learning method at scale, providing a complete robotic learning system.

## 3   Background

In RL, we learn policies $\pi$ that maximizes the expected total reward in a Markov decision process (MDP) with states $s$, actions $a$, discount factor $\gamma \in (0, 1]$, transition function $T(s'|s, a)$ and a reward function $R(s, a)$. Actions $a$ have dimensionality $d_{\mathcal{A}}$. Value-based RL approaches learn a Q-function $Q(s, a)$ representing the total discounted return $\sum_t \gamma^t R(s_t, a_t)$, with policy $\pi(a|s) = \arg\max_a Q(s, a)$. The Q-function can be learned by iteratively applying the Bellman operator [67]:

$$\mathcal{B}^* Q(s_t, a_t) = R(s_t, a_t) + \gamma \max_{a_{t+1}} Q(s_{t+1}, a_{t+1}),$$

approximated via function approximation and sampling. The offline RL setting assumes access to an offline dataset of transitions or episodes, produced by some unknown behavior policy $\pi_\beta(a|s)$, but does not assume the ability to perform additional online interaction during training. This is appealing

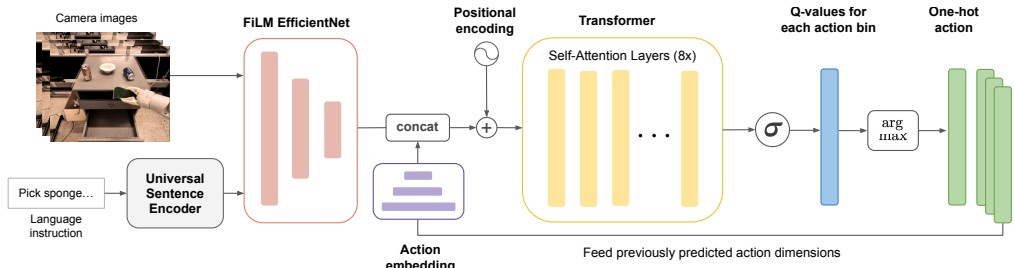

Figure 3: **Q-Transformer network architecture**, as applied to our multi-task language-conditioned robotic control setting. The encoding of the observations is concatenated with embeddings of the previous predicted action dimensions and processed by Transformer layers. We apply a sigmoid to the Transformer output to produce Q-values (normalized to lie in the range $[0, 1]$) for each of the action value bins. Finally, one-hot action vectors are constructed by taking the $\arg\max$ over all bins and are fed back to the network to predict the Q-values of the next action dimensions. The language instruction is encoded with *Universal Sentence Encoder* [68] and then fed to *FiLM EfficientNet* [69, 70] network together with the robot camera images.

for real-world robotic learning, where on-policy data collection is time-consuming. Learning from offline datasets requires addressing distributional shift, since in general the action that maximizes $Q(s_{t+1}, a_{t+1})$ might lie outside of the data distribution. One approach to mitigate this is to add a conservative penalty [29, 52] that pushes down the Q-values $Q(s, a)$ for any action $a$ outside of the dataset, thus ensuring that the maximum value action is in-distribution.

In this work, we consider tasks with sparse rewards, where a binary reward $R \in \{0, 1\}$ (indicating success or failure) is assigned at the last time step of episodes. Although our method is not specific to this setting, such reward structure is common in robotic manipulation tasks that either succeed or fail on each episode, and can be particularly challenging for RL due to the lack of reward shaping.

## 4 Q-Transformer

In this section, we introduce Q-Transformer, an architecture for offline Q-learning with Transformer models, which is based on three main ingredients. First, we describe how we apply discretization and autoregression to enable TD-learning with Transformer architectures. Next, we introduce a particular conservative Q-function regularizer that enables learning from offline datasets. Lastly, we show how Monte Carlo and $n$-step returns can be used to improve learning efficiency.

### 4.1 Autoregressive Discrete Q-Learning

Using Transformers with Q-learning presents two challenges: **(1)** we must tokenize the inputs to effectively apply attention mechanisms, which requires discretizing the action space; **(2)** we must perform maximization of Q-values over discretized actions while avoiding the curse of dimensionality. Addressing these issues within the standard Q-learning framework requires new modeling decisions. The intuition behind our autoregressive Q-learning update is to treat each action dimension as essentially a separate time step. That way, we can discretize individual dimensions (1D quantities), rather than the entire action space, avoiding the curse of dimensionality. This can be viewed as a simplified version of the scheme proposed in [65], though we apply this to high-capacity Transformer models, extend it to the offline RL setting, and scale it up to real-world robotic learning.

Let $\tau = (s_1, a_1, \ldots, s_T, a_T)$ be a trajectory of robotic experience of length $T$ from an offline dataset $\mathcal{D}$. For a given time-step $t$, and the corresponding action $a_t$ in the trajectory, we define a per-dimension view of the action $a_t$. Let $a_t^{1:i}$ denote the vector of action dimensions from the first dimension $a_t^1$ until the $i$-th dimension $a_t^i$, where $i$ can range from 1 to the total number of action dimensions, that we denote as $d_{\mathcal{A}}$. Then, for a time window $w$ of state history, we define the Q-value of the action $a_t^i$ in the $i-th$ dimension using an autoregressive Q-function conditioned on states from this time window $s_{t-w:t}$ and previous action dimensions for the current time step $a_t^{1:i-1}$. To train the Q-function, we define a per-dimension Bellman update. For all dimensions $i \in \{1, \ldots, d_{\mathcal{A}}\}$:

$$Q(s_{t-w:t}, a_t^{1:i-1}, a_t^i) \leftarrow \begin{cases} \max_{a_t^{i+1}} Q(s_{t-w:t}, a_t^{1:i}, a_t^{i+1}) & \text{if } i \in \{1, \ldots, d_{\mathcal{A}} - 1\} \\ R(s_t, a_t) + \gamma \max_{a_{t+1}^1} Q(s_{t-w+1:t+1}, a_{t+1}^1) & \text{if } i = d_{\mathcal{A}} \end{cases} \tag{1}$$

The reward is only applied on the last dimension (second line in the equation), as we do not receive any reward before executing the whole action. In addition, we only discount Q-values between the time steps and keep discounting at $1.0$ for all but the last dimension within each time step, to ensure the same discounting as in the original MDP. Figure 2 illustrates this process, where each yellow box represents the Q-target computation with additional conservatism and Monte Carlo returns described in the next subsections. It should be noted that by treating each action dimension as a time step for the Bellman update, we do not change the general optimization properties of Q-learning algorithms and the principle of the Bellman optimality still holds for a given MDP as we maximize over an action dimension given the optimality of all action dimensions in the future. We show that this approach provides a theoretically consistent way to optimize the original MDP in Appendix A, with a proof of convergence in the tabular setting in Appendix B.

## 4.2 Conservative Q-Learning with Transformers

Having defined a Bellman backup for running Q-learning with Transformers, we now develop a technique that enables learning from offline data, including human demonstrations and autonomously collected data. This typically requires addressing over-estimation due to the distributional shift, when the Q-function for the target value is queried at an action that differs from the one on which it was trained. Conservative Q-learning (CQL) [29] minimizes the Q-function on out-of-distribution actions, which can result in Q-values that are significantly smaller than the minimal possible cumulative reward that can be attained in any trajectory. When dealing with sparse rewards $R \in \{0, 1\}$, results in [27] show that the Q-function regularized with a standard conservative objective can take on negative values, even though instantaneous rewards are all non-negative. This section presents a modified version of conservative Q-learning that addresses this issue in our problem setting.

The key insight behind our design is that, rather than minimizing the Q-values on actions not in the data, we can instead regularize these Q-values to be close to the minimal attainable possible cumulative reward. Concretely, denoting the minimal possible reward on the task as $R_{\min}$, and the time horizon of the task as $T$, our approach regularizes the Q-values on actions not covered by the dataset towards $R_{\min} \cdot T$, which in our problem setting is equal to 0 (i.e., $R_{\min} = 0$). For simplicity of notation, we omit the action dimension indices in presenting the resulting objective, but remark that the training objective below is applied to Bellman backups on all action dimensions as described in the previous section. Let $\pi_\beta$ be the behavioral policy that induced a given dataset $\mathcal{D}$, and let $\tilde{\pi}_\beta(a|s) = \frac{1}{Z(s)} \cdot (1.0 - \pi_\beta(a|s))$ be the distribution over all actions which have a very low density under $\pi_\beta(a|s)$. Our objective to train the Q-function is:

$$ J = \frac{1}{2} \underbrace{\mathbb{E}_{s \sim \mathcal{D}, a \sim \pi_\beta(a|s)} \left[ \left( Q(s,a) - \mathcal{B}^* Q^k(s,a) \right)^2 \right]}_{(i), \text{ TD error}} + \alpha \cdot \frac{1}{2} \underbrace{\mathbb{E}_{s \sim \mathcal{D}, a \sim \tilde{\pi}_\beta(a|s)} \left[ \left( Q(s,a) - 0 \right)^2 \right]}_{(ii), \text{ conservative regularization } \mathcal{L}_C}, \quad (2) $$

where the first term $(i)$ trains the Q-function by minimizing the temporal difference error objective as defined in Eq. 1, and the second term $(ii)$ regularizes the Q-values to the minimal possible Q-value of 0 in expectation under the distribution of actions induced by $\tilde{\pi}_\beta$, which we denote as a conservative regularization term $\mathcal{L}_C$. Term $(ii)$ is also weighted by a multiplier $\alpha$, which modulates the strength of this conservative regularization. We discuss the choice of $\alpha$ in our implementation in Appendix D.2 and analyze the behavior of the conservatism term in Appendix C, providing a simple characterization of how this regularizer modifies the learned Q-function in tabular settings.

## 4.3 Improving Learning Efficiency with Monte Carlo and $n$-step Returns

When the dataset contains some good trajectories (e.g., demonstrations) and some suboptimal trajectories (e.g., autonomously collected trials), utilizing Monte Carlo return-to-go estimates to accelerate Q-learning can lead to significant performance improvements, as the Monte Carlo estimates along the better trajectories lead to much faster value propagation. This has also been observed in prior work [31]. Based on this observation, we propose a simple improvement to Q-Transformer that we found to be quite effective in practice. The Monte Carlo return is defined by the cumulative reward within the offline trajectory $\tau$: $\mathrm{MC}_{t:T} = \sum_{j=t}^{T} \gamma^{j-t} R(s_j, a_j)$. This matches the Q-value of the behavior policy $\pi_\beta$, and since the optimal $Q^*(s,a)$ is larger than the Q-value for any other policy, we have $Q^*(s_t, a_t) \geq \mathrm{MC}_{t:T}$. Since the Monte Carlo return is a lower bound of the optimal Q-function, we can augment the Bellman update to take the maximum between the MC-return and the current Q-value: $\max(\mathrm{MC}_{t:T}, Q(s_t, a_t))$, without changing what the Bellman update will converge to.

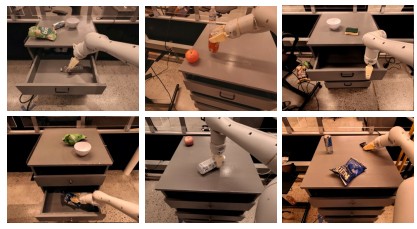

| Task category | # of tasks | Q-T | DT | IQL | RT-1 |
|---|---|---|---|---|---|
| drawer pick and place | 18 | 64% | 49% | 11 % | 17% |
| open and close drawer | 7 | 33% | 11% | 11 % | 0% |
| move object near target | 47 | 71% | 40% | 60% | 58% |
| Average success rate | | **56**% | 33% | 27% | 25% |

Figure 4: **Left**: Real world manipulation tasks. **Right:** Real world performance comparison. RT-1 [1] is imitation learning on demonstrations. Q-Transformer (Q-T), Decision Transformer (DT) [32], Implicit Q-learning (IQL) [40] learn from both demonstrations and autonomous data.

Although this does not change convergence, including this maximization speeds up learning (see Section 5.3). We present a hypothesis why this occurs. In practice, Q-values for final timesteps $(s_T, a_T)$ are learned first and then propagated backwards in future gradient steps. It can take multiple gradients for the Q-value to propagate all the way to $(s_1, a_1)$. The $\max(\text{MC}, Q)$ allows us to apply useful gradients to $Q(s_1, a_1)$ at the start of training before the Q-values have propagated.

In our experiments, we also notice that additionally employing $n$-step returns [71, 72] over action dimensions can significantly help with the learning speed. We pick $n$ such that the final Q-value of the last dimension of the next time step is used as the Q-target. This is because we get a new state and reward only after inferring and executing the whole action as opposed to parts of it, meaning that intermediate rewards remain 0 all the way until the last action dimension. While this introduces bias to the Bellman backups, as is always the case with off-policy learning with $n$-step returns, we find in our ablation study in Section 5.3 that the detrimental effects of this bias are small, while the speedup in training is significant. This is consistent with previously reported results [72]. More details about our Transformer sequence model architecture (depicted in Figure 3) conservative Q-learning implementation, and the robot system can be found in Appendix D.

## 5 Experiments

In our experiments, we aim to answer the following questions: (1) Can Q-Transformer learn from a combination of demonstrations and sub-optimal data? (2) How does Q-Transformer compare to other methods? (3) How important are the specific design choices in Q-Transformer? (4) Can Q-Transformer be applied to large-scale real world robotic manipulation problems?

### 5.1 Real-world language-conditioned manipulation evaluation

***Training dataset.*** The offline data used in our experiments was collected with a fleet of 13 robots, and consists of a subset of the demonstration data described by Brohan et al. [1], combined with lower quality autonomously collected data. The demonstrations were collected via human teleoperation for over 700 distinct tasks, each with a separate language description. We use a maximum of 100 demonstrations per task, for a total of about 38,000 demonstrations. All of these demonstrations succeed on their respective tasks and receive a reward of 1.0. The rest of the dataset was collected by running the robots autonomously, executing policies learned via behavioral cloning.

To ensure a fair comparison between Q-Transformer and imitation learning methods, we discard all *successful* episodes in the autonomously collected data when we train our method, to ensure that by including the autonomous data the Q-Transformer does not get to observe more successful trials than the imitation learning baselines. This leaves us with about 20,000 additional autonomously collected *failed* episodes, each with a reward of 0.0, for a dataset size of about 58,000 episodes. The episodes are on average 35 time steps in length. Examples of the tasks are shown in Figure 4.

***Performance evaluation.*** To evaluate how well Q-Transformer can perform when learning from real-world offline datasets while effectively incorporating autonomously collected failed episodes, we evaluate Q-Transformer on 72 unique manipulation tasks, and a variety of different skills, such as "drawer pick and place", "open and close drawer", "move object near target", each consisting of 18, 7 and 48 unique tasks instructions respectively to specify different object combinations and drawers. As such, the average success rate in Table 4 is the average over 72 tasks.

Since each task in the training set only has a maximum of 100 demonstrations, we observe from Figure 4 that an imitation learning algorithm like RT-1 [1], which also uses a similar Transformer architecture, struggles to obtain a good performance when learning from only the limited pool of

successful robot demonstrations. Existing offline RL methods, such as IQL [40] and a Transformer-based method such as Decision Transformer [32], can learn from both successful demonstrations and failed episodes, and show better performance compared to RT-1, though by a relatively small margin. Q-Transformer has the highest success rate and outperforms both the behavior cloning baseline (RT-1) and offline RL baselines (Decision Transformer, IQL), exceeding the average performance of the best-performing prior method by about 70%. This demonstrates that Q-Transformer can effectively improve upon human demonstrations using autonomously collected sub-optimal data.

Appendix G also shows that Q-Transformer can be successfully applied in combination with a recently proposed language task planner [8] to perform both affordance estimation and robot action execution. Q-Transformer outperforms prior methods for planning and executing long-horizon tasks.

## 5.2 Benchmarking in simulation

In this section, we evaluate Q-Transformer on a challenging simulated offline RL task that require incorporating sub-optimal data to solve the task. In particular, we use a visual simulated picking task depicted in Figure 5, where we have a small amount of position controlled human demonstrations (∼8% of the data). The demonstrations are replayed with noise to generate more trajectories (∼92% of the data). Figure 5 shows a comparison to several offline algorithms, such as QT-Opt with CQL [11, 29], IQL [40], AW-Opt [73], and Decision Transformer [32], along with RT-1 using Behavioral Cloning [1] on demonstrations only. As we see, algorithms that can effectively perform TD-learning to combine optimal and sub-optimal data (such as Q-Transformer and QT-Opt) perform better than others. BC with RT-1 is not

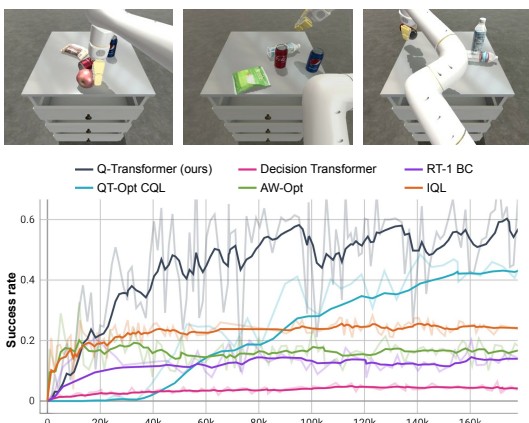

Figure 5: Performance comparison on a simulated picking task.

able to take advantage of sub-optimal data. Decision Transformer is trained on both demonstrations and sub-optimal data, but is not able to leverage the noisy data for policy improvement and does not end up performing as well as our method. Although IQL and AW-Opt perform TD-learning, the actor remains too close to the data and can not fully leverage the sub-optimal data. Q-Transformer is able to both bootstrap the policy from demonstrations and also quickly improve through propagating information with TD-learning. We also analyze the statistical significance of the results by training with multiple random seeds in Appendix F.

## 5.3 Ablations

We perform a series of ablations of our method design choices in simulation, with results presented in Figure 6 (left). First, we demonstrate that our choice of conservatism for Q-Transformer performs better than the standard CQL regularizer, which corresponds to a *softmax* layer on top of the Q-function outputs with a cross-entropy loss between the dataset action and the output of this softmax [29]. This regularizer plays a similar role to the one we propose, decreasing the Q-values for out-of-distribution actions and staying closer to the behavior policy.

As we see in Figure 6 (left), performance with softmax conservatism drops to around the fraction of demonstration episodes (∼8%). This suggests a collapse to the behavior policy as the conservatism penalty becomes too good at constraining to the behavior policy distribution. Due to the nature of the softmax, pushing Q-values down for unobserved actions also pushes Q-values up for the observed actions, and we theorize this makes it difficult to keep Q-values low for sub-optimal in-distribution actions that fail to achieve high reward. Next, we show that using conservatism is important. When removing conservatism entirely, we observe that performance collapses. Actions that are rare in the dataset will have overestimated Q-values, since they are not trained by the offline Q-learning procedure. The resulting overestimated values will propagate and collapse the entire Q-function, as described in prior work [38]. Finally, we ablate the Monte-Carlo returns and again observe performance collapse. This demonstrates that adding information about the sampled future returns significantly helps in bootstrapping the training of large architectures such as Transformers.

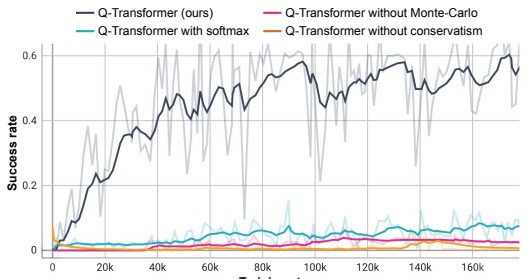

| $n$-step ablation | $n$-step | 1-step | 1-step |
|---|---|---|---|
| # of gradient steps | 137480 | 582960 | 136920 |
| Training duration (hours) | 32 | 163 | 40 |
| pick object | 94% | 97% | 92% |
| move object near target | 88% | 80% | 67% |

| Large offline dataset | Q-T | DT | RT-1 |
|---|---|---|---|
| Average success rate | **88%** | 78% | 82% |

Figure 6: **Left**: Ablations: changing to softmax conservatism decreases performance. Removing MC returns or conservatism completely collapse performance. **Top Right:** The $n$-step return version of our method reaches similar performance to the standard version with 4 times fewer steps, indicating that the added bias from $n$-step returns is small compared to the gain in training speed. Using $n$-step return also leads to better performance on tasks that have longer horizon, e.g. *move object near target*. **Bottom Right:** Success rates on real world task categories with a larger dataset.

We also ablate the choice of $n$-step returns from the Section 4.3 on real robots and observe that using $n$-step returns leads to a significantly faster training speed as measured by the number of gradient steps and wall clock time compared to using 1-step returns, with a minimal loss in performance, as shown in Figure 6 (top right).

### 5.4 Massively scaling up Q-Transformer

The experiments in the previous section used a large dataset that included successful demonstrations and failed autonomous trials, comparable in size to some of the largest prior experiments that utilized demonstration data [74, 15, 58]. We also carry out a preliminary experiment with a much larger dataset to investigate the performance of Q-Transformer as we scale up the dataset size.

This experiment includes all of the data collected with 13 robots and comprises of the demonstrations used by RT-1 [1] and successful autonomous episodes, corresponding to about 115,000 successful trials, and an additional 185,000 failed autonomous episodes, for a total dataset size of about 300,000 trials. Model architecture and hyperparameters were kept exactly the same, as the computational cost of the experiment made further hyperparameter tuning prohibitive (in fact, we only train the models once). Note that with this number of successful demonstrations, even standard imitation learning with the RT-1 architecture already performs very well, attaining 82% success rate. However, as shown in Figure 6 (bottom right), Q-Transformer was able to improve even on this very high number. This experiment demonstrates that Q-Transformer can continue to scale to extremely large dataset sizes, and continues to outperform both imitation learning with RT-1 and Decision Transformer.

## 6 Limitations and Discussion

In this paper, we introduced the Q-Transformer, an architecture for offline reinforcement learning with high-capacity Transformer models that is suitable for large-scale multi-task robotic RL. Our framework does have several limitations. First, we focus on sparse binary reward tasks corresponding to success or failure for each trial. While this setup is reasonable for a broad range of episodic robotic manipulation problems, it is not universal, and we expect that Q-Transformer could be extended to more general settings as well in the future.

Second, the per-dimension action discretization scheme that we employ may become more cumbersome in higher dimensions (e.g., controlling a humanoid robot), as the sequence length and inference time for our model increases with action dimensionality. Although $n$-step returns mitigate this to a degree, the length of the sequences still increases with action dimensionality. For such higher-dimensional action space, adaptive discretization methods might also be employed, for example by training a discrete autoencoder model and reducing representation dimensionality. Uniform action discretization can also pose problems for manipulation tasks that require a large range of motion granularities, e.g. both coarse and fine movements. In this case, adaptive discretization based on the distribution of actions could be used for representing both types of motions.

Finally, in this work we concentrated on the offline RL setting. However, extending Q-Transformer to online finetuning is an exciting direction for future work that would enable even more effective autonomous improvement of complex robotic policies.

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

# A  Proof of MDP optimization consistency

To show that transforming MDP into a per-action-dimension form still ensures optimization of the original MDP, we show that optimizing the Q-function for each action dimension is equivalent to optimizing the Q-function for the full action.

If we consider the full action $a_{1:d_\mathcal{A}}$ and that we switch to the state $s'$ at the next timestep, the Q-function for optimizing over the full action MDP would be:

$$\max_{a_{1:d_\mathcal{A}}} Q(s, a_{1:d_\mathcal{A}}) = \max_{a_{1:d_\mathcal{A}}} \left[ R(s, a_{1:d_\mathcal{A}}) + \gamma \max_{a_{1:d_\mathcal{A}}} Q(s', a_{1:d_\mathcal{A}}) \right]$$
$$= R(s, a^*_{1:d_\mathcal{A}}) + \gamma \max_{a_{1:d_\mathcal{A}}} Q(s', a_{1:d_\mathcal{A}}), \tag{3}$$

where $R(s, a^*_{1:d_\mathcal{A}})$ is the reward we get after executing the full action.

The optimization over each action dimension using our Bellman update is:

$$\max_{a_i} Q(s, a^*_{1:i-1}, a_i) = \max_{a_i} \mathcal{B}^* Q(s, a^*_{1:i-1}, a_i)$$
$$= \max_{a_i} \left[ \max_{a_{i+1}} Q(s, a^*_{1:i-1}, a_i, a_{i+1}) \right]$$
$$= \max_{a_i} \left[ \max_{a_{i+1}} \mathcal{B}^* Q(s, a^*_{1:i-1}, a_i, a_{i+1}) \right]$$
$$= \max_{a_i} \left[ \max_{a_{i+1}} \left( \max_{a_{i+2}} Q(s, a^*_{1:i-1}, a_i, a_{i+1}, a_{i+2}) \right) \right]$$
$$= R(s, a^*_{1:d_\mathcal{A}}) + \gamma \max_{a_1} Q(s', a_1)$$
$$= R(s, a^*_{1:d_\mathcal{A}}) + \gamma \max_{a_1} \mathcal{B}^* Q(s', a_1)$$
$$= R(s, a^*_{1:d_\mathcal{A}}) + \gamma \max_{a_1} \left[ \max_{a_2} Q(s', a_1, a_2) \right]$$
$$= R(s, a^*_{1:d_\mathcal{A}}) + \gamma \max_{a_1, a_2} [\mathcal{B}^* Q(s', a_1, a_2)]$$
$$= R(s, a^*_{1:d_\mathcal{A}}) + \gamma \max_{a_1, a_2} \left[ \max_{a_3} Q(s', a_1, a_2, a_3) \right]$$
$$= R(s, a^*_{1:d_\mathcal{A}}) + \gamma \max_{a_{1:d_\mathcal{A}}} Q(s', a_{1:d_\mathcal{A}}),$$

which optimizes the original full action MDP as in Eq. 3.

# B  Proof of convergence

Convergence of Q-learning has been shown in the past [67, 75]. Below we demonstrate that per-action dimension Q-function converges as well, by providing a proof almost identical to the standard Q-learning convergence proof, but extended to account for the per-action dimension maximization.

Let $d_\mathcal{A}$ be the dimensionality of the action space, $a$ indicates a possible sequence of actions, whose dimension is not necessarily equal to the dimension of the action space. That is:

$$a \in \{a_{1:i}, \forall i \leq d_\mathcal{A}\}$$

To proof convergence, we can demonstrate that the Bellman operator applied to the per-action dimension Q-function is a contraction, i.e.:

$$||\mathcal{B}^* Q_1(s, a) - \mathcal{B}^* Q_2(s, a)||_\infty \leq c ||Q_1(s, a) - Q_2(s, a)||_\infty,$$

where

$$\mathcal{B}^*Q(s,a) = \begin{cases} R(s,a) + \gamma \max_{a'} Q(s,a,a') & \text{if the dimension of } a \text{ is less than } d_{\mathcal{A}} \\ R(s,a) + \gamma \max_{a'} \underset{s'}{E}[Q(s',a')] & \text{if the dimension of } a \text{ is equal to } d_{\mathcal{A}} \end{cases}$$

$a'$ is the next action dimension following the sequence $a$, $s'$ is the next state of the MDP, $\gamma$ is the discounting factor, and $0 \le c \le 1$.

*Proof:* We can show that this is the case as follows:

Case 1: For action sequence whose dimension is less than the dimension of the action space.

$$\begin{aligned} &\mathcal{B}^*Q_1(s,a) - \mathcal{B}^*Q_2(s,a) \\ &= R(s,a) + \gamma \max_{a'} Q_1(s,a,a') - R(s,a) - \gamma \max_{a'} Q_2(s,a,a') \\ &= \gamma \max_{a'} [Q_1(s,a,a') - Q_2(s,a,a')] \\ &\le \gamma \sup_{s,a} [Q_1(s,a) - Q_2(s,a)] \\ \Longrightarrow &||\mathcal{B}^*Q_1(s,a) - \mathcal{B}^*Q_2(s,a)||_\infty \le \gamma ||Q_1(s,a) - Q_2(s,a)||_\infty \end{aligned}$$

where $\sup_{s,a}$ is the supremum over all action sequences, with $0 \le \gamma \le 1$ and $||f||_\infty = \sup_x[f(x)]$.

Case 2: For action sequence whose dimension is equal to the dimension of the action space

$$\begin{aligned} &\mathcal{B}^*Q_1(s,a) - \mathcal{B}^*Q_2(s,a) \\ &= R(s,a) + \gamma \max_{a'} \underset{s'}{E}[Q_1(s',a')] - R(s,a) - \gamma \max_{a'} \underset{s'}{E}[Q_2(s',a')] \\ &= \gamma \max_{a'} \underset{s'}{E}[Q_1(s',a') - Q_2(s',a')] \\ &\le \gamma \sup_{s,a}[Q_1(s,a) - Q_2(s,a)] \\ \Longrightarrow &||\mathcal{B}^*Q_1(s,a) - \mathcal{B}^*Q_2(s,a)||_\infty \le \gamma ||Q_1(s,a) - Q_2(s,a)||_\infty \end{aligned}$$

$\square$

## C  Analysis of the conservatism term

With the goal of understanding the behavior of our training procedure, we theoretically analyze the solution obtained by Eq. 2 for the simpler cases when $Q$ is represented as a table, and when the objective in Eq. 2 can be minimized exactly. We derive the minimizer of the objective in Eq. 2 by differentiating $J$ with respect to $Q$:

$$\begin{aligned} \forall s,a,k, \quad \frac{dJ}{dQ(s,a)} &= 0 \\ \pi_\beta(a|s)\left(Q(s,a) - \mathcal{B}^*Q^k(s,a)\right) + \alpha\tilde{\pi}_\beta(a|s)Q(s,a) &= 0 \\ Q(s,a)\left(\pi_\beta(a|s) + \alpha\tilde{\pi}_\beta(a|s)\right) &= \pi_\beta(a|s)\mathcal{B}^*Q^k(s,a) \\ Q^{k+1}(s,a) = \underbrace{\frac{\pi_\beta(a|s)}{\pi_\beta(a|s) + \alpha\tilde{\pi}_\beta(a|s)}}_{:=m(s,a)} \cdot \mathcal{B}^*Q^k(s,a) & \end{aligned} \qquad (4)$$

Eq. 4 implies that training with the objective in Eq. 2 performs a weighted Bellman backup: unlike the standard Bellman backup, training with Eq. 2 multiplies large Q-value targets by a weight $m(s,a)$. This weight $m(s,a)$ takes values between 0 and 1, with larger values close to 1 for in-distribution actions where $(s,a) \in \mathcal{D}$, and very small values close to 0 for out-of-distribution actions $a$ at any state $s$ (i.e., actions where $\pi_\beta(a|s)$ is small). Thus, the Bellman backup induced via Eq. 4 should effectively prevent over-estimation of Q-values for unseen actions.

# D  Q-Transformer Architecture & System

In this section, we describe the architecture of Q-Transformer as well as the important implementation and system details that make it an effective Q-learning algorithm for real robots.

## D.1  Transformer sequence model architecture

Our neural network architecture is shown in Figure 3. The architecture is derived from the RT-1 design [1], adapted to accommodate the Q-Transformer framework, and consists of a Transformer backbone that reads in images via a convolutional encoder followed by tokenization. Since we apply Q-Transformer to a multi-task robotic manipulation problem where each task is specified by a natural language instruction, we first embed the natural language instruction into an embedding vector via the Universal Sentence Encoder [68]. The embedding vector and images from the robot camera are then converted into a sequence of input tokens via a FiLM EfficientNet [69, 70]. In the standard RT-1 architecture [1], the robot action space is discretized and the Transformer sequence model outputs the logits for the discrete action bins per dimension and per time step. In this work, we extend the network architecture to use Q-learning by applying a sigmoid activation to the output values for each action, and interpreting the resulting output after the sigmoid as Q-values. This representation is particularly suitable for tasks with sparse per-episode rewards $R \in [0, 1]$, since the Q-values may be interpreted as probabilities of task success and should always lie in the range $[0, 1]$. Note that unlike the standard softmax, this interpretation of Q-values does *not* prescribe normalizing *across* actions (i.e., *each* action output can take on any value in $[0, 1]$).

Since our robotic system, described in Section D.3, has 8-dimensional actions, we end up with 8 dimensions per time step and discretize each one into $N = 256$ value bins. Our reward function is a sparse reward that assigns value 1.0 at the last step of an episode if the episode is successful and 0.0 otherwise. We use a discount rate $\gamma = 0.98$. As is common in deep RL, we use a target network to estimate target Q-values $Q^k$, using an exponential moving average of $Q$-network weights to update the target network. The averaging constant is set to 0.01.

## D.2  Conservative Q-learning implementation

The conservatism penalty in Section 4.2 requires estimating expectations under $\pi_\beta(a|s)$ and $\tilde{\pi}_\beta(a|s) \propto (1-\pi_\beta(a|s))$, with the latter being especially non-trivial to estimate. We employ a simple and crude approximation that we found to work well in practice, replacing $\pi_\beta(a|s)$ with the empirical distribution corresponding, for each sampled state-action tuple $(s_j, a_j) \in \mathcal{D}$, to a Dirac delta centered on $a_j$, such that $\pi_\beta(a|s_j) = \delta(a = a_j)$. This results in a simple expression for $\tilde{\pi}_\beta(a|s_j)$ corresponding to the uniform distribution over all *other* actions, such that $\tilde{\pi}_\beta(a|s_j) \propto \delta(a \neq a_j)$. After discretizing the actions, there are $N - 1$ bins per dimension to exhaustively iterate over when computing the conservatism term in Eq. 2, which is the same as taking the average over targets for all unseen action values. In our experiments, we find that simply setting the conservatism weight to $\alpha = 1.0$ worked best, without additional tuning.

## D.3  Robot system overview

The robot that we use in this work is a mobile manipulator with a 7-DOF arm with a 2 jaw parallel gripper, attached to a mobile base with a head-mounted RGB camera, illustrated in Figure 1. The RGB camera provides a $640 \times 512$ RGB image, which is downsampled to $320 \times 256$ before being consumed by the Q-Transformer. See Figure 4 for images from the robot camera view. The learned policy is set up to control the arm and the gripper of the robot. Our action space consists of 8 dimensions: 3D position, 3D orientation, gripper closure command, and an additional dimension indicating whether the episode should terminate, which the policy must trigger to receive a positive reward upon successful task completion. Position and orientation are relative to the current pose, while the gripper command is the absolute closedness fraction, ranging from fully open to fully closed. Orientation is represented via axis-angles, and all actions except whether to terminate are continuous actions discretized over their full action range in 256 bins. The termination action is binary, but we pad it to be the same size as the other action dimensions to avoid any issues with unequal weights. The policy operates at 3 Hz, with actions executed asynchronously [76].

---

**Algorithm 1** Temporal difference error and loss computation for one action dimension $i$ at timestep $t$, $a_t^i$.

---

**Input** Sequence of state in time window of size $w$, $s_{t-w:t}$.
**Input** Language embedding of task instruction $l$.
**Input** The state at timestep $t + 1$, $s_{t+1}$.
**Input** Dataset action up to dimension $i$, $\{_\mathcal{D} a_t^j\}_{j=0}^i$.
**Output** The loss to optimize Q-Transformer.

$Q^{targ} \leftarrow$ Compute maximum Q-values of the next action dimension using Eq. 1

```
// Compute the maximum between Q-target and Monte Carlo return.
```
$Q^{targ} \leftarrow \max(\text{MC}, Q^{targ})$

```
// Compute the temporal difference error.
```
$\text{TDError} = \dfrac{1}{2} \left( \text{Q-Transformer}(l, s_{t-w:t}, \{a^j\}_{j=1}^i) - Q^{targ} \right)^2$

```
// Compute the conservative regularizer.
// The sum is over all action bins not equal to the tokenized
   dataset action.
// N is the number of discretization bin.
```
$\text{Reg} = \dfrac{1}{2(N-1)} \sum_{a \neq _\mathcal{D} a_t^i} \left( \text{Q-Transformer}(l, s_{t-w:t}, \{a^j\}_{j=1}^{i-1} \cup \{a\}) \right)^2$

```
// Compute the loss function
```
$\mathcal{L} = \text{TDError} + \text{Reg}$

Return $\mathcal{L}$ as the loss function to optimize Q-Transformer with.

---

## E   Pseudo-code

Algorithm 1 shows the loss computation for training each action dimension of the Q-Transformer. We first use Eq. 1 to compute the maximum Q-values over the next action dimensions. Then we compute the Q-target for the given dataset action by using the Bellman update with an additional maximization over the Monte-Carlo return and predicted maximum Q-value at the next time step. The TD-error is then computed using the Mean-Squared Error. Finally, we set a target of 0 for all discretized action bins except the dataset action and add the averaged Mean-Squared Error over these dimensions to the TD-Error, which results in the total loss $\mathcal{L}$.

## F   Running training for multiple random seeds

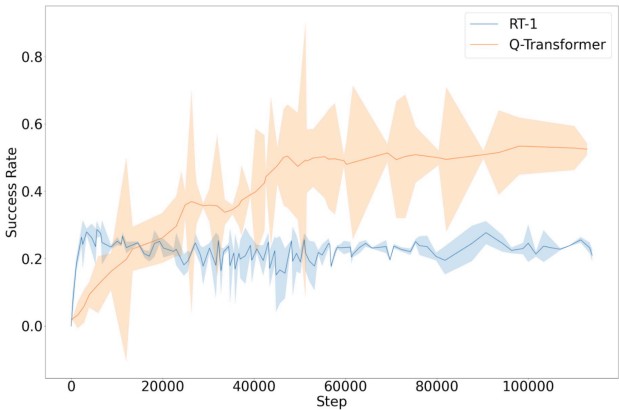

Figure 7: Mean and variance of Q-Transformer and RT-1 performance in simulation when running the training for 5 different random seeds.

In addition to performing a large amount of evaluations, we also analyze the statistical significance of our learning results by running our training of Q-Transformer and RT-1 on multiple seeds in simulation. In particular, we run the training for 5 random seeds in Figure 7. As we can see, Q-Transformer retains its improved performance across the distribution of the random seeds.

## G   Q-Transformer value function with a language planner experiments

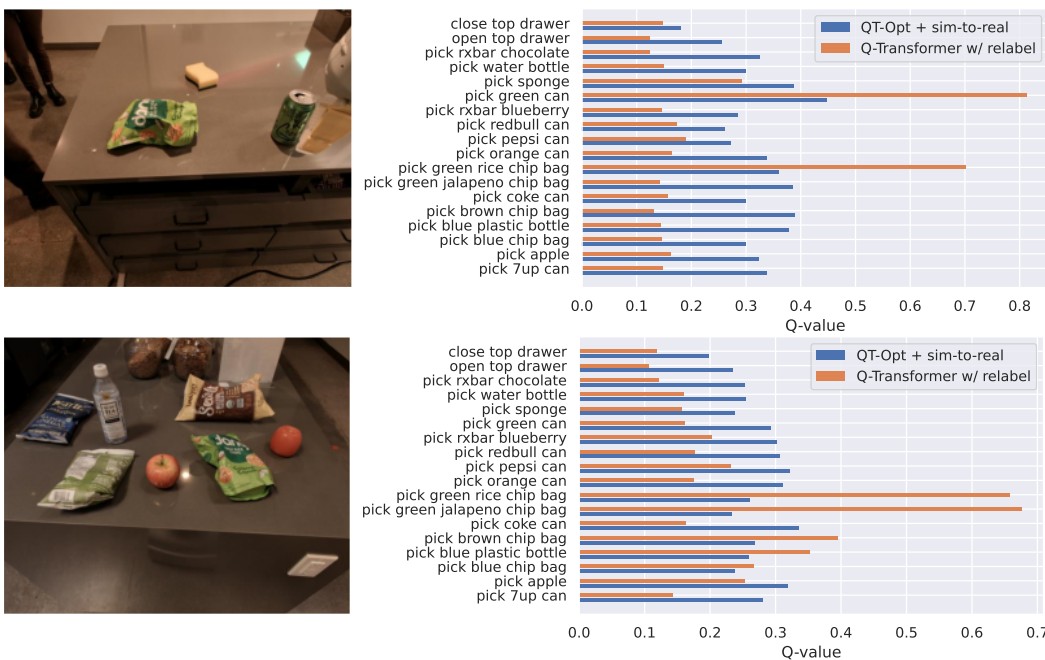

Figure 8: Qualitative comparisons of Q-values from QT-Opt (sim-to-real) and Q-Transformer. Q-Transformer outputs sharper Q-values for objects close to the robot, which can be grasped faster and more easily than the objects farther away.

Recently, the SayCan algorithm [8] was proposed as a way to combine large language models (LLMs) with learned policies and value functions to solve long-horizon tasks. In this framework, the value function for each available skill is used to determine the "affordance" of the current state

for that skill, and a large language model then selects from among the available affordances to take a step towards performing some temporally extended task. For example, if the robot is commanded to bring all the items on a table, the LLM might propose a variety of semantically meaningful items, and select from among them based on the item grasping skill that currently has a high value (corresponding to items that the robot thinks it can grasp). SayCan uses QT-Opt in combination with sim-to-real transfer to train Q-functions for these affordances. In the following set of experiments, we demonstrate that the Q-Transformer outperforms QT-Opt for affordance estimation without using any sim-to-real transfer, entirely using the real world dataset that we employ in the preceding experiments.

We first benchmark Q-Transformer on the problem of correctly estimating task affordances from the RT-1 dataset [1]. In addition to the standard training on demonstrations and autonomous data, we introduce a training with relabeling, which we found particularly useful for affordance estimation. During relabeling, we sample a random alternate task for a given episode. We relabel the task name of the episode to the newly sampled task, and set reward to 0.0. This ensures that the boundaries between tasks are more clearly learned during train-

| Model | Precision | Recall | F1 |
|---|---|---|---|
| QT-Opt (sim-to-real) | 0.61 | 0.68 | 0.64 |
| Q-T w/ relabel | 0.76 | 0.89 | **0.82** |
| Q-T w/o relabel | 0.58 | 0.93 | 0.71 |

Table 1: Affordance estimation comparison: precision, recall and F1 score when using Q-values to determine if a task is feasible. Q-Transformer (Q-T) with multi-task relabeling consistently produces better affordance estimates.

ing. Table 1 shows comparison of performance of our model with and without relabeling as well as the sim-to-real QT-Opt model used in SayCan [8]. Both of our models outperform the QT-Opt model on F1 score, with the relabeled model outperforming it by a large margin. This demonstrates that our Q-function can be effectively used for affordance estimation, even without training with sim-to-real transfer. Visualization of the Q-values produced by our Q-function can be found in Figure 8.

We then use Q-Transformer in a long horizon SayCan style evaluation, replacing both the sim-to-real QT-Opt model for affordance estimation, and the RT-1 policy for low-level robotic control. During this evaluation, a PaLM language model [77] is used to propose task candidates given a user query. Q-values are then used to pick the task candidate with the highest affordance score, which is then executed on the robot using the execution policy. The Q-Transformer used for affordance estimation is trained with relabeling. The Q-Transformer used for low-level control is

| Method | | Success Rate | |
|---|---|---|---|
| Affordance | Execution | Planning | Execution |
| Q-T w/ relabel | Q-T | **93** | **93** |
| QT-Opt (sim-to-real) | RT-1 | 87 | 67 |

Table 2: Performance on SayCan style long-horizon tasks: SayCan queries $Q(s, a)$ in planning to pick a language instruction, then runs a policy to execute the plan. Q-Transformer outperforms RT-1 with QT-Opt in both planning and execution.

trained without relabeling, since we found relabeling episodes at the task level did not improve execution performance. SayCan with Q-Transformer is better at both planning the sequence of tasks and executing those plans, as illustrated in Table 2.

## H Real robotic manipulation tasks used in our evaluation

We include the complete list of evaluation tasks in our real robot experiments below.

**Drawer pick and place**: pick 7up can from top drawer and place on counter, place 7up can into top drawer, pick brown chip bag from top drawer and place on counter, place brown chip bag into top drawer, pick orange can from top drawer and place on counter, place orange can into top drawer, pick coke can from middle drawer and place on counter, place coke can into middle drawer, pick orange from middle drawer and place on counter, place orange into middle drawer, pick green rice chip bag from middle drawer and place on counter, place green rice chip bag into middle drawer, pick blue plastic bottle from bottom drawer and place on counter, place blue plastic bottle into bottom drawer, pick water bottle from bottom drawer and place on counter, place water bottle into bottom drawer,

pick rxbar blueberry from bottom drawer and place on counter, place rxbar blueberry into bottom drawer.

**Open and close drawer**: open top drawer, close top drawer, open middle drawer, close middle drawer, open bottom drawer, close bottom drawer.

**Move object near target**: move 7up can near apple, move 7up can near blue chip bag, move apple near blue chip bag, move apple near 7up can, move blue chip bag near 7up can, move blue chip bag near apple, move blue plastic bottle near pepsi can, move blue plastic bottle near orange, move pepsi can near orange, move pepsi can near blue plastic bottle, move orange near blue plastic bottle, move orange near pepsi can, move redbull can near rxbar blueberry, move redbull can near water bottle, move rxbar blueberry near water bottle, move rxbar blueberry near redbull can, move water bottle near redbull can, move water bottle near rxbar blueberry, move brown chip bag near coke can, move brown chip bag near green can, move coke can near green can, move coke can near brown chip bag, move green can near brown chip bag, move green can near coke can, move green jalapeno chip bag near green rice chip bag, move green jalapeno chip bag near orange can, move green rice chip bag near orange can, move green rice chip bag near green jalapeno chip bag, move orange can near green jalapeno chip bag, move orange can near green rice chip bag, move redbull can near sponge, move sponge near water bottle, move sponge near redbull can, move water bottle near sponge, move 7up can near blue blastic bottle, move 7up can near green can, move blue plastic bottle near green can, move blue plastic bottle near 7up can, move green can near 7up can, move green can near blue plastic bottle, move apple near brown chip bag, move apple near green jalapeno chip bag, move brown chip bag near green jalapeno chip bag, move brown chip bag near apple, move green jalapeno chip bag near apple, move green jalapeno chip bag near brown chip bag.

