# OpenReview forum: "Q-Transformer: Scalable Offline Reinforcement Learning via Autoregressive Q-Functions"
_robot-learning.org/CoRL/2023/Conference — CoRL 2023 Poster_

### Official Review · Reviewer_VHvc · 2023-07-01

**Confidence:** 4
**Originality:** Good
**Technical Quality:** Fair
**Clarity Of Presentation:** Good
**Impact:** 2

**Recommendation:**

Weak Accept: I recommend accepting the paper, but will not argue for my recommendation if the majority of other reviewers have a different opinion.

**Review:**

I have reviewed this paper already in the version submitted to RSS and this version seems substantially identical. As for the previous version, I am positive about the quality of writing, presentation, and potential significance of this work. Experimental results are also convincing. However, I am still concerned about the soundness of the action-value maximization. In my opinion, optimizing each dimension separately does not lead to a joint optimal action. In other words, given $a_1^*=argmax_{a_1}Q(s_{1:2},a_1)$ and $a_2^*=argmax_{a_2}Q(s_{1:2},a_1^*,a_2)$, I do not think that $(a_1^*,a_2^*)=argmax_{a_1,a_2}Q(s_{1:2},a_1,a_2)$ holds. The reason is that $a_1^*$ is found without considering the other dimension $a_2$, which could affect the optimality of action $a_1^*$. The paper states that

> It should be noted that by treating each action dimension as a time step for
168 the Bellman update, we do not change the general optimization properties of Q-learning algorithms
169 and the principle of the Bellman optimality still holds for a given MDP as we maximize over an
170 action dimension given the optimality of all action dimensions in the future.

however, this claim should be supported by a rigorous proof. Even if I consider the paper of good quality, I have to express a very negative overall evaluation due to this lack of mathematical soundness. In case a valid proof will be provided, I'd be willing to increase my score.

**Quality Of The Limitations Section:**

Limitations are addressed clearly

**Questions For Rebuttal:**

- Can you formally prove the correctness of your maximization scheme over each dimension of the action space?

**Robotics Focus:**

Sufficient demonstration on hardware

**Summary Of Paper:**

This paper introduces a method to enable the use of transformers for learning action-value functions in reinforcement learning. The proposed method, called Q-transformer, discretize each dimension of the action space and optimizes each dimension separately. The method additionally leverages conservative Q-learning for dealing with offline data. Experimental results are positive in challenging high-dimensional real-world manipulation tasks.

**Summary Of Recommendation:**

The paper is well written and the proposed method is potentially a significant contribution to enabling the use of transformers in reinforcement learning. However, the lack of soundness of the proposed method is a critical issue that forces me to provide a very negative overall score. As stated before, in case the authors will provide a valid formal proof to support their claims, I'd be willing to increase my score.

---

> ### Comment · Reviewer_9Xry · 2023-08-03
> **Action decomposition**
>
> I (fellow reviewer) would like to gently push back on the following claim from this review:
>
> "In my opinion, optimizing each dimension separately does not lead to a joint optimal action."
>
> Acting according to the max-Q in each dimension in sequence leads to an optimal joint action as long as each subsequent dimension takes the previous dimensions into account as extra state variables, which is definitely the case here due to the transformer architecture.  You would be correct if the Q function for each dimension were computed unconditionally based only on the current MDP state/POMDP history, but as long as the function computing the Q value for a_n depends on the selection of a_0...a_n-1, everything works out fine.
>
> In other words, what the authors are doing is modelling this and acting greedily, which works fine:
> Q(a_0|s), Q(a_1|s,a_0), Q(a_2|s,a_0,a_1)...
>
> But you would be correct that modelling this and acting greedily does not:
> Q(a_0|s), Q(a_1|s), Q(a_2|s)...
>
> This has been covered pretty well by prior work, for example "Discrete Sequential Prediction of Continuous Actions For Deep RL" [Metz 2017].  It may be true that this paper would be better served by covering these details, or at least mentioning the potential confusion and providing a clarifying citation to prior work, but I would gently suggest that this should be a minor concern and not a major stumbling block.  The method is sound.

---

> > ### Comment · Reviewer_VHvc · 2023-08-08
> > **Response to Action Decomposition**
> >
> > Thank you. Please check the response to the authors.

---

> ### Author Response · Authors · 2023-08-04
> **Proof of action maximization**
>
> Thank you for your review. The maximization over the action dimensions corresponds to a straightforward dynamic programming approach for computing a max over the action, where the max over each dimension of the action is computed as a function of preceding dimensions. It is straightforward to show that this is equivalent to standard Q-learning, and we present this argument below (which we will add to the paper).
>
> As mentioned in our paper, we employ a per-action-dimension Bellman update (Eq. 1 in. the paper) meaning that Bellman optimality holds for training the model to select an overall optimal action that takes into account the optimality of future action dimension. To consider a concrete example of selecting the first two dimensions of an action (assuming action dimensionality $d_\mathcal{A} > 3$), given that we do not have intermediate rewards and do not discount between action dimensions, the Q-value maximization for a joint optimization would be:
> \begin{align*}
> \max_{a_1, a_2} Q(s, a_1, a_2) = \max_{a_1, a_2} \mathcal{B}^* Q(s, a_1, a_2) = \max_{a_1, a_2} \left [\max_{a_3} Q(s, a_1, a_2, a_3)\right].
> \end{align*}
>
> With our procedure of applying the Bellman update per-dimension we can show that maximization over the first action dimension $a_1$ leads to the same maximized Q-value:
> \begin{align*}
> \max_{a_1} Q(s, a_1) = \max_{a_1} \mathcal{B}^* Q(s, a_1)
> \end{align*}
> \begin{align*}
> =\max_{a_1} \left [\max_{a_2} Q(s, a_1, a_2)\right] \end{align*}
> \begin{align*}
> =\max_{a_1} \left [\max_{a_2} \mathcal{B}^* Q(s, a_1, a_2)\right]
> \end{align*}
> \begin{align*}
> =  \max_{a_1} \left [\max_{a_2} \left(\max_{a_3} Q(s, a_1, a_2, a_3)\right)\right]
> \end{align*}
> \begin{align*}
> = \max_{a_1, a_2} \left [\max_{a_3} Q(s, a_1, a_2, a_3)\right].
> \end{align*}
>
> More generally, if we consider the full action $a_{1:d_\mathcal{A}}$ and that we switch to the state $s'$ at the next timestep, the optimization over the whole action would be:
> \begin{align*}
> \max_{a_{1:d_\mathcal{A}}} Q(s, a_{1:d_\mathcal{A}}) &= \max_{a_{1:d_\mathcal{A}}} \left [R(s,a_{1:d_\mathcal{A}}) + \gamma \max_{a_{1}} Q(s', a_1)\right] = R(s,a_{1:d_\mathcal{A}}^*) + \gamma \max_{a_{1}} Q(s', a_1),
> \end{align*}
> where $R(s,a_{1:d_\mathcal{A}}^*)$ is the reward we get after executing the whole action.
>
>
> The optimization over each action dimension in the general case is:
> \begin{align*}
> \max_{a_i} Q(s, a_{1:i-1}^*, a_i) = \max_{a_i} \mathcal{B}^* Q(s, a_{1:i-1}^*, a_i)
> \end{align*}
> \begin{align*}
> = \max_{a_i} \left [\max_{a_{i+1}} Q(s, a_{1:i-1}^*, a_i, a_{i+1})\right]
> \end{align*}
> \begin{align*}
> = \max_{a_i} \left [\max_{a_{i+1}} \mathcal{B}^* Q(s, a_{1:i-1}^*, a_i, a_{i+1})\right]
> \end{align*}
> \begin{align*}
> = \max_{a_i} \left [\max_{a_{i+1}} \left( \max_{a_{i+2}} Q(s, a_{1:i-1}^*, a_i, a_{i+1}, a_{i+2})\right)\right]
> \end{align*}
> \begin{align*}
> = R(s,a_{1:d_\mathcal{A}}^*) + \gamma \max_{a_{1}} Q(s', a_1)
> \end{align*}
> where we keep on expanding the recursion through the Bellman operator until the last dimension. This demonstrates that optimization over each action dimension optimizes the overall Q-value of the whole action.
>
> In our paper, we also describe how $n$-step returns  can circumvent optimization over all action dimensions and that although introducing bias they can help to speed up learning on our tasks as shown in Section 5.3. This technique is complimentary and does not change our presented method.

---

> > ### Comment · Reviewer_VHvc · 2023-08-08
> > **Response to Proof of action maximization**
> >
> > Thank you for your response. The proof makes sense to me but still does not clear my concerns. In particular, as cited in the paper and also pointed out by a reviewer, I checked the paper "Discrete Sequential Prediction of Continuous Actions For Deep RL" [Metz 2017], from which the proposed action decomposition scheme is inspired. However, this paper only strengthens my doubts. [Metz 2017] proposes a hierarchical approach where multi-dimensional actions belonging to the original high-level MDP are decomposed into 1-D actions to control a low-level MDP. [Metz 2017] proposes to learn each 1-D action by applying an update based on the optimal Bellman operator using the next 1-D action and reward 0 till the final 1-D action to form the whole multi-dimensional action of the high-level MDP. This makes sense to me, although I think that maximizing over a different action-space at each step calls for a formal proof of convergence which is missing in [Metz 2017]. Nevertheless, [Metz 2017] recognizes that this optimization scheme would not work without a way to maintain consistency between the action-values of the low-level and high-level MDPs. To achieve this, they add a consistent term in the loss (Equation 8). To me, it seems like this consistency term is crucial for the effectiveness of the maximization scheme over decomposed actions.
> >
> > I'd appreciate it if the authors could clear this doubt and explain why they did not address/discuss the consistency of the action-values.

---

> > > ### Author Response · Authors · 2023-08-08
> > > **MDP Consistency**
> > >
> > > We note that in our method, the TD backup on the last action dimension of each time step already serves a similar function as the "consistency" loss in the Metz et al. work.
> > >
> > > To see that the above presented proof ensures consistency, we can continue on expanding the Bellman operator:
> > > \begin{align*}
> > > \max_{a_i} Q(s, a_{1:i-1}^*, a_i) = R(s,a_{1:d_\mathcal{A}}^*) + \gamma \max_{a_{1}} Q(s', a_1)
> > > \end{align*}
> > > \begin{align*}
> > >  = R(s,a_{1:d_\mathcal{A}}^*) + \gamma \max_{a_{1}} \mathcal{B}^* Q(s', a_1)
> > > \end{align*}
> > > \begin{align*}
> > >  = R(s,a_{1:d_\mathcal{A}}^*) + \gamma \max_{a_{1}} \left [ \max_{a_2} Q(s', a_1, a_2) \right]
> > > \end{align*}
> > > \begin{align*}
> > >  = R(s,a_{1:d_\mathcal{A}}^*) + \gamma \max_{a_1, a_2} \left [ \mathcal{B}^* Q(s', a_1, a_2) \right]
> > > \end{align*}
> > > \begin{align*}
> > >  = R(s,a_{1:d_\mathcal{A}}^*) + \gamma \max_{a_1, a_2} \left [ \max_{a_3} Q(s', a_1, a_2, a_3) \right]
> > > \end{align*}
> > > \begin{align*}
> > >  = R(s,a_{1:d_\mathcal{A}}^*) + \gamma \max_{a_{1:d_{\mathcal{A}}}}  Q(s', a_{1:d_{\mathcal{A}}}),
> > > \end{align*}
> > > which optimizes the original full action MDP.

---

> > > > ### Author Response · Authors · 2023-08-11
> > > > **Please let us know if our response addresses your concerns**
> > > >
> > > > Please let us know if our response addresses your concerns, or if you still have any concerns about the correctness of the method.

---

> > > > > ### Comment · Reviewer_VHvc · 2023-08-12
> > > > > **Doubts about the proof**
> > > > >
> > > > > Thank you for the additional proof. However, I am still not fully convinced. The proof seems to simply show an equivalence between $Q*$ and $\mathcal{B}Q*$, using the decomposed actions, given the optimal $Q$ function. However, it does not show that the proposed method is sound to find the optimal action-value function $Q*$. To me, this proof is similar to proving that $Q$-Learning converges to $Q*$ by showing that $Q*=\mathcal{B}Q*$, which is obviously not sufficient.

---

> > > > > > ### Author Response · Authors · 2023-08-13
> > > > > > **Proof that the Bellman operator when applied to the per-action dimension Q-function is a contraction**
> > > > > >
> > > > > > Q-Learning convergence is a well-established result in the literature. We have showed that the Bellman update for decomposed actions effectively results in the Bellman update for the whole actions. Below we demonstrate that per-action dimension Q-function converges as well, by providing a proof almost identical to the standard Q-learning convergence proof, but extended to account for the per-action dimension maximization.
> > > > > >
> > > > > > Let $|\mathcal{A}|$ be the action space, $a$ indicates a possible sequence of action, whose dimension is not necessarily equal to the dimension of the action space. That is:
> > > > > >
> > > > > > \begin{align*} a \in \\{ a_{1:i}, \forall i \leq |\mathcal{A}| \\} \end{align*}
> > > > > >
> > > > > > The Bellman operator applied to the per-action dimension Q-function is a contraction, i.e.:
> > > > > >
> > > > > > \begin{align*}    || B^ * Q_1(s, a) - B^* Q_2(s, a)|| \leq c||Q_1(s, a) - Q_2(s, a)||, \end{align*}
> > > > > >
> > > > > > where:
> > > > > >
> > > > > > \begin{equation}
> > > > > >     \mathcal{B^*} Q(s, a ) = \begin{cases}
> > > > > >        R(s, a ) + \gamma \hspace{0.3em} \underset{a'}{\max} \hspace{0.3em} Q(s, a, a') & \text{if the dimension of $a$ is less than } |\mathcal{A}| \\\\
> > > > > >       R(s, a ) + \gamma \hspace{0.3em} \underset{a'}{\max} \hspace{0.3em} \underset{s'}{E} \left[ Q(s', a') \right] &  \text{if the dimension of $a$ is equal to } |\mathcal{A}|
> > > > > >     \end{cases}
> > > > > > \end{equation}
> > > > > >
> > > > > > $a'$ is the next action dimension following the sequence $a$, $s'$ is the next state of the MDP, $\gamma$ is the discounting factor, and $0\leq c\leq 1$.
> > > > > >
> > > > > > Proof: We can show that this is the case as follows:
> > > > > >
> > > > > > Case 1: for action sequences whose dimension is less than the dimension of the action space.
> > > > > >
> > > > > > \begin{align*}  B^* Q_1 (s, a) - B^* Q_2(s, a)  \end{align*}
> > > > > >
> > > > > > \begin{align*}  = R(s, a) + \gamma\max_{a'} Q_1(s, a, a') - R(s, a) - \gamma\max_{a'} Q_2( s, a, a' )  \end{align*}
> > > > > >
> > > > > > \begin{align*}  = \gamma\max_{ a' } \left[ Q_1( s, a, a' ) - Q_2( s, a, a' ) \right] \end{align*}
> > > > > >
> > > > > > \begin{align*}  \leq \gamma \sup_{ s, a } [Q_1( s, a ) -Q_2( s, a )] \end{align*}
> > > > > >
> > > > > > \begin{align*}  \Longrightarrow || B^ * Q_1(s, a) - B^* Q_2(s, a)|| \leq \gamma ||Q_1(s, a) - Q_2(s, a)||  \end{align*}
> > > > > >
> > > > > > where the $\sup_{ s, a }$ is the supremum over all action sequences, with $0\leq \gamma \leq 1$ and $||f|| = \sup_{x}[f(x)]$.
> > > > > >
> > > > > > Case 2: for action sequence whose dimension is equal to the dimension of the action space
> > > > > >
> > > > > > \begin{align*}  B^* Q_1 (s, a) - B^* Q_2(s, a)  \end{align*}
> > > > > >
> > > > > > \begin{align*}  = R(s, a ) + \gamma \max_{a'} \underset{s'}{E} [ Q_1(s', a') ] - R(s, a) - \gamma \max_{a'} \underset{s'}{E} [ Q_2(s', a') ]  \end{align*}
> > > > > >
> > > > > > \begin{align*}  = \gamma\max_{ a' } \underset{s'}{E} \left[ Q_1( s', a' ) - Q_2( s', a' ) \right]  \end{align*}
> > > > > >
> > > > > > \begin{align*}  \leq \gamma \sup_{ s, a } [Q_1( s, a ) -Q_2( s, a )]  \end{align*}
> > > > > >
> > > > > > \begin{align*}  \Longrightarrow || B^* Q_1(s, a) - B^* Q_2(s, a)|| \leq \gamma ||Q_1(s, a) - Q_2(s, a)||  \end{align*}
> > > > > >
> > > > > > End of proof.

---

> > > > > > > ### Comment · Reviewer_VHvc · 2023-08-15
> > > > > > > **The method seems sound now**
> > > > > > >
> > > > > > > I really thank the Authors and Reviewer 9Xry for providing proofs about the soundness of the method and engaging into discussion. This helped me to clear my doubts about the soundness of the method, and now I am confident enough about its correctness. However, I think that the paper needs to include part of the analysis conducted during this discussion phase, especially considering the counter-intuitive (at least to me) nature of the proposed method. Simply saying that it is similar to Metz et al. is not sufficient, because after all it is not so similar as we have all agreed.
> > > > > > >
> > > > > > > Nevertheless, I am happy to increase the score to a weak accept, which is not strong for the absence of the theoretical analysis of the method in the current version.

---

> > > > > > ### Comment · Reviewer_9Xry · 2023-08-14
> > > > > > **The result is sound.**
> > > > > >
> > > > > > The authors have presented a formal proof, which sufficiently shows that the discretization method works.  It is sound 100%, confidence level 5.  I say this as someone who is familiar with the related literature and has been looking into similar methods for some time.
> > > > > >
> > > > > > I jumped in early and tried to back up the authors' claims, providing the reference to Metz, but unfortunately, it looks like I might have muddied the waters to some extent, due to confusion about some additional things that Metz et al. do which is not relevant to the work presented here.  I apologize for any confusion I may have caused, and will come back to this at the end.
> > > > > >
> > > > > > I would like to present a less formal, but perhaps more intuitive argument to supplement the authors' proof, but before I do, I would humbly request that if this argument is not satisfying to VHvc, that this not be a cause for further speculation or doubts about the authors' approach and their earlier response.  Their proof is sufficient.  This is only meant to supplement.  This is my attempt at clarification, not theirs.  If it does not adequately make sense, please disregard it.  Here goes:
> > > > > >
> > > > > > We are considering two MDPs.  The first MDP has the original state and N-dimensional action space.  The second "expanded" MDP breaks each step in the original MDP into N steps, where each step is responsible for selecting the i-th component of an action that will be passed back to the original MDP.  As an important side note, the states in the expanded MDP contain a state from the original MDP, plus the list of 1 dimensional actions taken so far, so that when selecting action component a3, the model knows what has been chosen for a0, a1 and a2. The first thing to notice about these two MDPs is that there is a straightforward conversion of actions from the original MDP to N-step sequences of actions in the expanded MDP (due to the deterministic nature of the transitions that happen when accumulating action dimensions), and a conversion from these N-step sequences of actions in the expanded MDP back to single actions in the original MDP.
> > > > > >
> > > > > > The absolutely crucial thing to realize is that this conversion does not affect the expected returns or the expected state visitation frequencies.  If I gave you a state in the original MDP and a single action, the distribution of reward and next states you would expect to see is identical to the distribution you would see if you converted the action to an N-step sequence and passed it to the second MDP and vice versa.  The reason this is so important is that it shows that any policy that improves performance in either of these MDPs can be used to produce improved performance in the other.  Similarly optimal performance in one can be converted to optimal performance in the other.  Any algorithm (Q-learning or anything else) that improves the performance of a policy acting in one of these MDPs automatically leads to an improved policy in the other by simply going through the conversion process, so all you need to do is improve whichever one is most convenient for your to deal with.
> > > > > >
> > > > > > With respect to Metz et al., I think the confusion around equation 8 comes from the fact that they are trying to model the Q function in both the original MDP and the expanded MDP, and keep them consistent.  This is a design decision that they made to try to improve stability (since working in the expanded MDP increases the horizon and thus reduces the stability), but it is not necessary.  You can operate directly in the expanded MDP and everything works out fine.

---

### Official Review · Reviewer_5C59 · 2023-07-15

**Confidence:** 5
**Originality:** Good
**Technical Quality:** Good
**Clarity Of Presentation:** Good
**Impact:** 4

**Recommendation:**

Weak Accept: I recommend accepting the paper, but will not argue for my recommendation if the majority of other reviewers have a different opinion.

**Review:**

In my opinion this work is interesting, in the right direction but with limited impact. On the up side, there are 3 good points to this paper. First, it based on two input modals which is a new standard in robotics: language and images. Second, the system architecture has some advantages in comparison to previous work. It is based on both Film EfficientNet and a Q-Transformer where using the output of the network, the chosen action is propagated back to the Q-Transformer input in a closed loop fashion. Third, we note that the technical challenge that the Q-Transformer is solving is manifested in the three principles implemented by the Q-Transformer, namely, the discretization, conservative Q-function, and the usage of the Monte-Carlo n-step returns for improving the efficiency. These principles are proven to solve the tasks in hand.

On the down side, I argue that the experiments that the authors provide are not really 700, but of much lower scale. Actually, as the authors themselves provide in the appendix, it is roughly can be classified into 3 types: drawer pick and place, open and close drawer, and move object near target. In addition, I argue that what the authors really want to say is that their system can generalize over this three types. I value these experiments and the variability of in the text prompts, but I argue that this is expected outcome in this case.

Quality
The Quality of paper is satisfactory. First, in the chosen topic of how to use transformer in order to have a closed loop controller that can implement the architecture. Second, the considerations provided are logical. Specifically, I find the authors methodology (of disassembling the action space into its dimensions and training on each dimension separately) smart, specially when the experiments validate this although I'm not sure that for 6 degrees of continuous movement the alternative method of discretizing in these 6 dimensions would achieve less.

Clarity
The paper is written clearly although many real robot details are missing.

Originality
The work is build on foundations suggested in other domains such as the Monte-Carlo n-step backup, using the EfficientNet, Universal Sentence Encoder, adopting the Metz et al. scheme of discrete sequential prediction of continuous actions for deep RL.  On the other hand, as far as I know, some of the techniques are original as the architecture.

Strengths
To summarize: the strengths of the paper are the architecture, using SOTA components, demonstrate the method in experiments, the usage of both vision and language for operating the robot, and finally, make all these mentioned things work together.

Weaknesses
The experiments are limited and some design choices are not grounded enough. I must say that I know how hard is to make things work so I'm not position that as a must be, but rather a "I wish it was explored more profoundly".

**Quality Of The Limitations Section:**

Limitations are addressed clearly

**Questions For Rebuttal:**

(1) Do you have results regarding not splitting the actions by the axes? I would appreciate see how much gain can one have in this context.
(2) Can you please show the technique performance on contact reach tasks such as insertion? Such tasks are more "closed loop" in high frequency tasks that I suspect the given technique cannot tackle.
(3) What are the actual system performance in terms of:
 - Latency
 - Cycle time
 - Compute needed in inference time?
(4) What is the SW stack used in the experiments? ROS? What is the motion planner?


**Robotics Focus:**

Sufficient demonstration on hardware

**Summary Of Paper:**

The authors present "Q-Transformer" which is multi-task algorithm based on offline data. It leverages both human demonstrations and autonomous collected data. The authors argue that discretizing the action space naively leads to exponential blowup in the size of the action space. As a result, the authors propose to discretizing each action dimension separately and treat it in different time step. This enables the authors to use a a simple Q-learning method (unlike QT-Opt for example). In addition they use a regularizer that minimizes values of actions not appearing in the off-line data so they can handle offline RL distributional shift phenomenon. Lastly, they use Monte-Carlo N-step returns with TD that is shown to assist in perfrormance. The experiments are applied on 700 tasks where the training was done on 38K successful demonstrations and 20K failed autonomously collected episodes. These experiments show that the Q-transformer beats previous robotic-RL approaches.

**Summary Of Recommendation:**

I thank the authors for the work! I tend to accept the paper, but conditioned on that the authors will provide more details as requested in "questions for rebuttal".

---

### Official Review · Reviewer_9Xry · 2023-07-19

**Confidence:** 4
**Originality:** Very Good
**Technical Quality:** Very Good
**Clarity Of Presentation:** Very Good
**Impact:** 4

**Recommendation:**

Strong Accept: I recommend accepting the paper and will argue for my recommendation even if other reviewers hold a different opinion.

**Review:**

Pros:
+ The system performs well on real-world robotics problems
+ The method has high practical utility
+ The paper is written clearly and is easy to understand
+ The paper contains many practical modifications to prior work that may aid further development.

Cons:
- Figure 6 indicates that the system requires all of the modifications presented here in order to function, and that removing any of them dramatically hurts performance.  This may indicate some brittleness in the system and hyperparameters and is worth exploring further.
- (MINOR) The experiments cover robotic pick-and-place tasks, which is definitely sufficient for this paper, however it would be worth exploring these ideas in other domains to determine how general these improvements are.

**Quality Of The Limitations Section:**

Limitations are addressed clearly

**Questions For Rebuttal:**

None at this time.

**Robotics Focus:**

Sufficient demonstration on hardware

**Summary Of Paper:**

The authors present a transformer-based discretized Q-learning method for offline reinforcement learning.  Their model improves over several offline RL baselines by significant margins.  They present several practical modifications necessary to achieve these results.

**Summary Of Recommendation:**

The authors method is practical and interesting with possibly future applications in a wide array of robotic control problems.  The experiments show strong performance on challenging problems.  This passes an important test: "would I want to read this paper when starting a project in this domain?" The answer is yes.

Post rebuttal: I liked this paper before and still like it.  Seems like some other authors' issues have been resolved.  Still strong accept.

---

### Official Review · Reviewer_SvFm · 2023-07-20

**Confidence:** 4
**Originality:** Good
**Technical Quality:** Good
**Clarity Of Presentation:** Very Good
**Impact:** 4

**Recommendation:**

Weak Accept: I recommend accepting the paper, but will not argue for my recommendation if the majority of other reviewers have a different opinion.

**Review:**

### Strengths

1. Overall, I found this paper to be generally well-written and well-organized and communicates its ideas clearly.
2. Transformers have the potential to provide significant gains in RL, much as they have in other sequence modeling tasks, but this aspect of their potential is still relatively under-explored. The steps this paper takes to make transformer architectures compatible with more standard RL optimization pipelines may be an important step in enabling more such research.
3. I appreciate that the paper is able to abstract away enough of the specifics of transformers to be able to have a meaningful discussion on how they may be integrated into RL, without getting bogged down in specifics, as I feel that that may have been distracting for this type of work.

### Weaknesses

1. ~~Given that we are dealing with RL problems, the fact that models are trained only once leads me to question the repeatability of the results.~~ How reliably can one expect performance gains - and what are the margins on performance differences with larger datasets (would, for example, the gap be narrowed between Q-T vs RT-1 with more statistical significance?). **UPDATE**: Given the added experimentation, I am less concerned about repeatability. The analysis is not as thorough as it could be, but is sufficient for me to not consider it a major limiting weakness.

2. ~~As a choice of baseline, I find that the decision transformer (DT) makes for a poor choice relative to existing work - specifically, if one were to do a web search for "Q learning transformer RL," one of the top hits is "Q-learning Decision Transformer: Leveraging Dynamic Programming for Conditional Sequence Modelling in Offline RL" by Yamagata et al. from 2022, which also addresses the limitations of DTs by augmenting training with RL. Given that the space of transformers applied directly to RL is relatively small, the omission seems particularly stark (especially given how, that paper was the top hit for searches on multiple search engines, is almost a year old, and seems highly relevant to the work in this paper). All that said,~~ I recognize the differences between the Q-transformer and QDT developed by Yamagata et al. and am not suggesting a lack of novelty in this work - merely that I feel an important point of discussion has been left out. **UPDATE**: I maintain that the paper would benefit from more discussion of existing work on QDTs (which the authors have committed to including), but I no longer regard it a weakness as strongly as I previously did.

3. ~~I find that the way lines are plotted in Figs. 5 and 6 can be misleading. It would seem that the training curves are being smoothed for clarity, which is in itself fine, but I would propose sticking to either the raw curves or the smoothed curves (with indications that they have been smoothed) - instead of potentially leading readers to believe that the darker lines may represent more meaningful aggregated statistics (like from multiple runs).~~

### Post-rebuttal notes:

With the additional experiments and clarifications, I believe the quality of the paper is improved. While there are still points that could be improved, I think the authors have done enough to cross the threshold into having an acceptable paper.

**Quality Of The Limitations Section:**

Additional details required

**Questions For Rebuttal:**

1. ~~How would you address the points raised in the 'Weaknesses' section in my review?~~

2. ~~Could you clarify what $a^0$ is in eq 1? Is the a vector not $d_A$-dimensional? It appeared that it was 1-indexed and was defined for $i \in [1, d_A]$ and yet a 0th index appears in eq. 1.~~

3. ~~Based on the results in Fig. 6, it would appear that both the innovations proposed here - the monte-carlo lower bound and the approach to conservatism - are important. However, it is curious that the softmax conservatism works so poorly by comparison. Could you provide some more intuition for this? With just one test, I would require convincing that this is not an issue with implementation but rather a fundamental limitation of the baseline approach.~~

4. ~~The table on the top right of Fig 6 has 2 columns labeled 1-step. What is the difference between them? I am not sure I am parsing this clearly.~~

### Post-rebuttal update:

All questions sufficiently answered.

**Robotics Focus:**

Sufficient demonstration on hardware

**Summary Of Paper:**

The paper presents Q-transformers, a transformer-based representation of Q networks designed to improve the performance of high-capacity models on multi-task offline RL problems when learning from large datasets. The proposed method employs a number of adjustments to the typical RL optimization structure to make it compatible with transformer architectures and the need for discrete tokens. The presented results indicate that, for more modest training datasets, performance gains over previous transformer-based supervised learning techniques can be significant, and also provides evidence to suggest that these gains also translate to improved performance when working with large datasets.

**Summary Of Recommendation:**

I think, overall, this is a good paper  but that it is limited by its lack of statistical evaluation. ~~While I understand the computational limitation, with just a single run's worth of experimental data, I remain skeptical, particularly given that RL performance typically invites skepticism. I also maintain that I believe the paper would benefit from more direct contrasting with QDTs - ideally experimentally, but at least with some meaningful discussion. For now, I will mark this as a weak reject but am willing to raise my score if other reviewers are accepting of the limited scope of experimentation and do not raise other critical concerns.~~

With the added experimentation and clarifications, I believe the paper is sufficiently improved for me to raise my score to a 'weak accept'.

---

### Decision · Program_Chairs · 2023-08-30

**Decision:**

Accept (Poster)

**Comment:**

This paper is one of the first approaches using large-scale transformers in combination with offline reinforcement learning to learn complex tasks from a set of demonstrations. The approach shows considerable improvements w.r.t. the presented baselines.

The rebuttal was extremely beneficial to this paper, the discussion between the authors and the reviewers clarify the soundness of the methods, solving the most problematic concerns of the reviewers. All reviewers now share a positive opinion of the paper, therefore I recommend accepting this work.

However, most reviewers, believe that this paper still has some shortcomings. The main concern is the reproducibility of the work, that in its current status appears to be quite limited. Also, during the discussion, it was evident that the method is mostly composed of minor incremental advances. While these components are important for deploying the method in complex environments and allowing effective learning of transformers-based policies, it is probably not enough for an oral presentation.

Indeed, the current score is close, but not above, the 93% percentile, and no reviewer has argued in favor of an oral presentation, despite my requests to provide support arguments in favor of this decision. Therefore, I suggest a poster presentation for this work.